# Implementation fidelity of tuberculosis preventive therapy for under five children exposed to sputum smear positive pulmonary tuberculosis in Kaski district, Nepal: An implementation research

**Ashmita Ghimire**[1]☯*, **Yodi Mahendradhata**[2]☯, **Sagun Paudel**[1], **Chhoden Lama Yonzon**[1], **Bhuvan K. C.**[3], **Sushmita Sharma**[4], **Adi Utarini**[2]☯

1 International Master Program in Public Health, Faculty of Medicine, Public Health and Nursing, Universitas Gadjah Mada, Yogyakarta, Indonesia, 2 Department of Health Policy and Management, Faculty of Medicine, Public Health and Nursing, Universitas Gadjah Mada, Yogyakarta, Indonesia, 3 School of Pharmacy, Monash University Malaysia, Selangor, Malaysia, 4 Province Health Logistic Management Centre, Gandaki Province, Pokhara, Nepal

☯ These authors contributed equally to this work.
* ashmi.gr8@gmail.com

**Data Availability Statement:** All relevant data are within the paper and its Supporting Information

## Abstract

### Introduction

In line with the WHO recommendation, Nepal has started implementing Tuberculosis prevention therapy (TBPT) for under five children exposed to Sputum Smear Positive Pulmonary Tuberculosis, as one of the strategies for prevention, care and control of TB. Implementation fidelity study is important to assess on what extent preventive program is being implemented. The objective of the study measured the implementation fidelity of TBPT program Kaski district, Nepal.

### Methods

We used a mixed-method explanatory sequential design study. Quantitative data were collected through retrospective review of records from April 2018 to May 2019 and level of adherence was established. Moderating factors influencing the implementation of TBPT were TBPT were assessed qualitatively. Sixteen in-depth interviews and a focus group discussion was conducted purposively with responsible stakeholders. The study was guided by the Conceptual Framework for Implementation Fidelity (CFIR) developed by Carroll.

### Results

The majority of the components of the TBPT program were found to be implemented with a moderate level of fidelity. The proportion of under five years children initiate and complete the full course of TBPT was 72.5% and 75.86% respectively. The proportion of index cases traced for household contact, contact tracing within two months and timely initiation of

files. Raw data and transcript cannot be shared publicly due to confidentiality issue involving personal detail of human subject. In order to request access to the data, please contact Emilia Sri Wulandari, emilia@ugm.ac.id, or Yuyun Yohana, yuyun.yohana@ugm.ac.id.

**Funding:** This research was a part of postgraduate thesis. The scholarship for this degree was funded by the Special Programme for Research and Training in Tropical Diseases at the World Health Organization (WHO/TDR) and Universitas Gadjah Mada, Indonesia.The funders had no role in study design, data collection and analysis, decision to publish, or preparation of the manuscript. The funder provided support in the form of salaries for authors [insert relevant initials], but did not have any additional role in the study design, data collection and analysis, decision to publish, or preparation of the manuscript. The specific roles of these authors are articulated in the 'author contributions' section.

**Competing interests:** The authors have declared that no competing interests exist.

therapy within two months were 54.19%, 82.73% and 86.20%. Moderating factors identified in the implementation of the program were contact tracing and enrollment, partnership and ownership, training resources, medication, awareness and information dissemination.

## Conclusion

The TBPT program is being moderately implemented in Kaski districts. Addressing the key challenges identified in contact tracing, partnership/ownership, incentives, training and knowledge of health workers results in more identification of children eligible for TBPT.

## Introduction

Tuberculosis (TB) is one of the top ten leading causes of death worldwide. In 2018, there were an estimated 10 million reported cases of TB globally, amongst which 1.1 million were children [1, 2]. Children under 5 years of age contribute an estimated 10%-20% burden of TB in endemic areas [3]. Southeast Asia and the Western Pacific region constitute a higher number of childhood TB, which included more than half of total TB cases in children, followed by the African region with around one-third of total childhood TB cases [1]. It is a significant contributor to morbidity and mortality in children under five years [4]. Patients with Sputum Smear positive (SS+) pulmonary TB (index cases) have a high rate of transmission of *Mycobacterium tuberculosis* to their children who encounter them [5]. Infected children are at greater risk of progression to TB than adults [6]. There is a double risk of childhood mortality if SS + cases belong to the same family and by eight folds when the mother had TB [7].

The World Health Organization (WHO) firmly states the need to treat latent TB infection in people living with HIV and children below 5 years who are in household contact with pulmonary bacteriologically confirmed (PBC) cases [1]. According to the current recommendation, six months of Isoniazid monotherapy (10 mg/kg/day, range 7–15 mg/kg, maximum dose 300 mg/day) is used to treat latent TB infection. However, the WHO Guidelines Development Group recommends a 3 months' daily Rifampicin plus Isoniazid for infants and children <15 years of age based on the ground that its benefits outweigh the harm, has a better safety profile and higher completion rate compared to Isoniazid monotherapy [8]. According to the data reported from the program implementation setting, Isoniazid preventive therapy (IPT) initiation and completion ranged from 22–64.3% and 80.3–90.90%, respectively [9, 10]. Likewise, IPT initiation and completion in research setting vary from 86.9–89.4% and 50.51–94.5% [11, 12]. Despite the WHO recommendation, several underlying challenges such as poor compliance, drug and dosage form unavailability, poor monitoring and insufficient knowledge among healthcare worker exists in implementing preventive therapy [13–16].

Nepal has rolled out six months of IPT for children under five years of age who are in close contact with PBC tuberculosis from the year 2017 [17]. It has also endorsed the new regimen of rifampicin and isoniazid daily for three months by replacing six months of isoniazid monotherapy from April 2019. The IPT programme in Nepal is implemented via Sub Receipent (SR) and health facility-based approaches. A sub receipent is an organization engaged by primary receipent (PR) at sub national level to carry out programme activities that are part of a Global Fund grant [18]. As per the WHO report, around half of the children eligible are not enrolled in preventive therapy in Nepal. The implementation fidelity of the programme has not been assessed [19]. Therefore, this study assesses the implementation fidelity of

Tuberculosis preventive therapy for under-five children exposed to sputum smear-positive pulmonary Tuberculosis in Nepal.

## Material and methods

### Study setting

We carried out the study in the Kaski district of Gandaki Province, Nepal. Kaski district is a medium TB burden districts in Gandaki Province and has medium notified PBC TB cases. The selection of the Kaski district was purposive as it matched our study objective. We selected six different health facilities that were providing TBPT during the given period for In-depth interviews (IDIs) and a Focus group discussion (FGD). Those selected health facilities included one Regional TB center (RTC), one district hospital (Shisuwa hospital), one urban health center (urban health center 17), two health posts (Hemja and Naudada health post), one urban DOTS center (Urban health promotion center 8). All of the selected health facilities were Directly Observed Treatment Short-course (DOTS) center.

### Study design

We used a mixed-method explanatory sequential design, which involved a quantitative study followed by a qualitative study. This study followed the standard for reporting the implementation studies (STARI) checklist.

**A conceptual framework for the study.**    We used the implementation fidelity framework proposed by Carroll et al. [20]. Implementation fidelity is defined as the degree to which an intervention/program is implemented as it is proposed or as recommended in the approved protocol [20]. This framework consists of five major components to be measured: adherence, comprehensiveness of policy, strategies to facilitate implementation, quality of delivery and patient responsiveness [20, 21].

Program adherence refers to the degree to which program components are delivered as prescribed by the model and includes the subcategories of content, frequency, duration and coverage [20–22]. Complexities of policy describe whether an intervention is vague or easy enough to be implemented or understood. Facilitation strategies are likely to optimize and standardize the fidelity [20]. Participant responsiveness measures the degree to which extent the participants are accepting and enthusiastic about the new interventions delivered by the providers [20].

### Data collection and analysis

Based on the framework, we run a simple descriptive test to analyze the adherence to the TBPT program. We did a retrospective review of data from April 2018 to May 2019. Besides, we explored moderating factors in terms of comprehensiveness of policy, facilitation strategies and patient responsiveness.

**Assessment of adherence.**    We designed a secondary data collection checklist (S1 File) and collected data through the tuberculosis preventive therapy register (TBPT), contact tracing register and TB master register from the district. Some data which were not available in the register and we took those data from the sub-recipient SR organization providing technical assistant to this program. We retrieved the data on the programmatic indicator of TBPT such as initiation/not initiation and completion of the full course of TBPT, contact tracing, time of contact tracing and enrollment in TBPT. We double entered the data into MS Excel and double clean it to minimize the error that could occur while retrieving information from the different registers. We then transferred the data to STATA 13 software for analysis. We did a

descriptive analysis to establish frequencies and percentages. The level of adherence was calculated for individual components. The proportion of individual indicator which falls within the range of 0–50% was categorized as low implemented, 51–79 were categorized as moderately implemented and 80–100% were categorized as highly implemented [23].

**Assessment of moderating factors.** We used an open structured questionnaire to assess moderating factors. We purposively selected the participants for the qualitative study and carried out sixteen IDIs, which included health care providers (n = 6), NGO representatives (n = 1), TB focal person (n = 1) and parents or caretakers of contact children (n = 8). We did one focus group discussion (FGD) with eight contact tracers. We developed the questionnaire based on the childhood contact case management algorithm at the health facility and community level, other components mentioned in the guideline and based on different literature reviews. We added additional questions that were raised in the first few interviews with the interview guide for future interviews. We recorded the interviews using a digital audio recorder. We did the interviews in the Nepali Language. We transcribed the IDIs and FGD recording and field notes on the same day of the interview to ensure the credibility and reliability of information gathered. And they were translated into English. We developed a list of open code, theme and organized them under the possible moderating factors. To ensure the validity, we reviewed data and code within the research team, then the difference in understanding was resolved via discussion until consensus was maintained.

**Data integration.** We gave equal weight to data from the various sources. The research team merged it at the analysis of different stages using the Implementation fidelity framework. We used qualitative themes to explain quantitative adherence data.

### Ethical approval

Ethical approval for this study was obtained from the Nepal Health Research Council (NHRC), Nepal (Reg no. 224 /2019). Permission was taken from the National Tuberculosis Center (NTC). Written consent was taken from the individual participants involved in qualitative research. Children were not included in the interview as information was gathered from parents/caretakers on behalf of children Anonymity and confidentiality of the individual participants were maintained.

## Results

### Data sources

A total of 310 SS+ pulmonary tuberculosis cases were registered between April 1, 2018 to May 31, 2019 at Regional TB center in Kaski district. The mean age ± standard deviation (SD) of registered patients was 41.99 ± 1.13 years with majority 170 (54.84%) from *Janajati* ethnic group. Detail description of sociodemographic and clinical characteristics of registered TB cases is shown in Table 1. Of the 310 total PBC cases, 168 (54.19%) were screened for household contacts. Out of screened, 139 (82.74%) were contact traced within two months (intensive phase) and rest 29 (17.26)% were screened after two months. Of the 40 children eligible for therapy only 29 (72.5%) children were enrolled in therapy. Twenty-two children (76%) already completed the course of therapy, seven (24%) were under therapy and there were no reported cases of withdrawal (Fig 1). Of the 310 total PBC cases, 168 (54.19%) were screened for household contacts. Out of screened, 139 (82.74%) were contact traced within two months (intensive phase) and rest 29 (17.26)% were screened after two months. Of the 40 children eligible for therapy only 29 (72.5%) children were enrolled in therapy. Twenty-two children (76%) already completed the course of therapy, seven (24%) were under therapy and there were no reported cases of withdrawal (Fig 1).

**Table 1. Sociodemographic and clinical profile of PBC tuberculosis and children enrolled in TBPT Kaski district.**

| Variables | Category | Registered PBC Tuberculosis cases | | Children enrolled in Tuberculosis preventive therapy | |
|---|---|---|---|---|---|
| | | Number (n) | Percentage (%) | Number (n) | Percentage (%) |
| **Total** | | 310 | 100 | **(29)** | 100 |
| **Gender** | Male | 196 | 63.23 | 9 | 31.03 |
| | Female | 114 | 36.77 | 20 | 69.96 |
| **Age** | 0–1 | - | - | 4 | 13.80 |
| | 1–2 | - | - | 12 | 41.37 |
| | >2 | - | - | 13 | 44.88 |
| **Age** | 0–14 | 9 | 2.90 | - | - |
| | 15–24 | 68 | 21.94 | - | - |
| | 25–34 | 53 | 17.10 | - | - |
| | 35–44 | 53 | 17.10 | - | - |
| | 45–54 | 38 | 12.26 | - | - |
| | 55–64 | 35 | 11.29 | - | - |
| | >65 | 54 | 17.42 | - | - |
| **Ethnicity** | *Dalit* | 53 | 17.10 | 9 | 31.03 |
| | *Janajati* | 170 | 54.84 | 10 | 34.50 |
| | *Madeshi* | 5 | 1.61 | 1 | 3.44 |
| | *Muslim* | 2 | 0.65 | - | - |
| | *Brahmin/Chhetri* | 80 | 25.80 | 9 | 31.03 |
| **TB Category** | New | 245 | 79.03 | - | - |
| | Relapse | 47 | 15.16 | - | - |
| | Treatment after lost to follow up | 3 | 0.97 | - | - |
| | Treatment after failure | 3 | 0.97 | - | - |
| | Previously treated history unknown | 2 | 0.65 | - | - |
| | Transfer In | 10 | 3.23 | 1 | 3.44 |
| **HIV Status** | Yes | 8 | 2.58 | 28 | 96.55 |
| | No | 164 | 52.90 | - | - |
| | Not documented | 138 | 44.52 | - | - |
| **Relation with the Index case** | Father | - | - | 8 | 27.59 |
| | Mother | - | - | 12 | 41.38 |
| | Grandparent | - | - | 7 | 24.14 |
| | Other | | | 2 | 6.90 |
| **The gap in TB registration and enrolment in therapy** | < Within two months | - | - | 25 | 86.20 |
| | >2months (3 and 4 months) | - | - | 4 | 13.80 |

Among a total of 29 children in therapy, 25 (82.74%) were enrolled within two months after the registration of index cases. Children's enrollment was also recorded in three and four months (Table 2). The factor associated with non-enrollment in therapy is described qualitatively (Fig 2). Likewise, 3 out of 5 components of the TBPT program were found to be implemented with a moderate level of fidelity (Table 2).

## Moderating factors

**A. Comprehensiveness of policy.** The concept of TBPT was introduce in Nepal as an approach for prevention of TB among children and treatment of latent TB. The TB prevention programs, despite having effective implementation policies, have not been conducted satisfactorily.

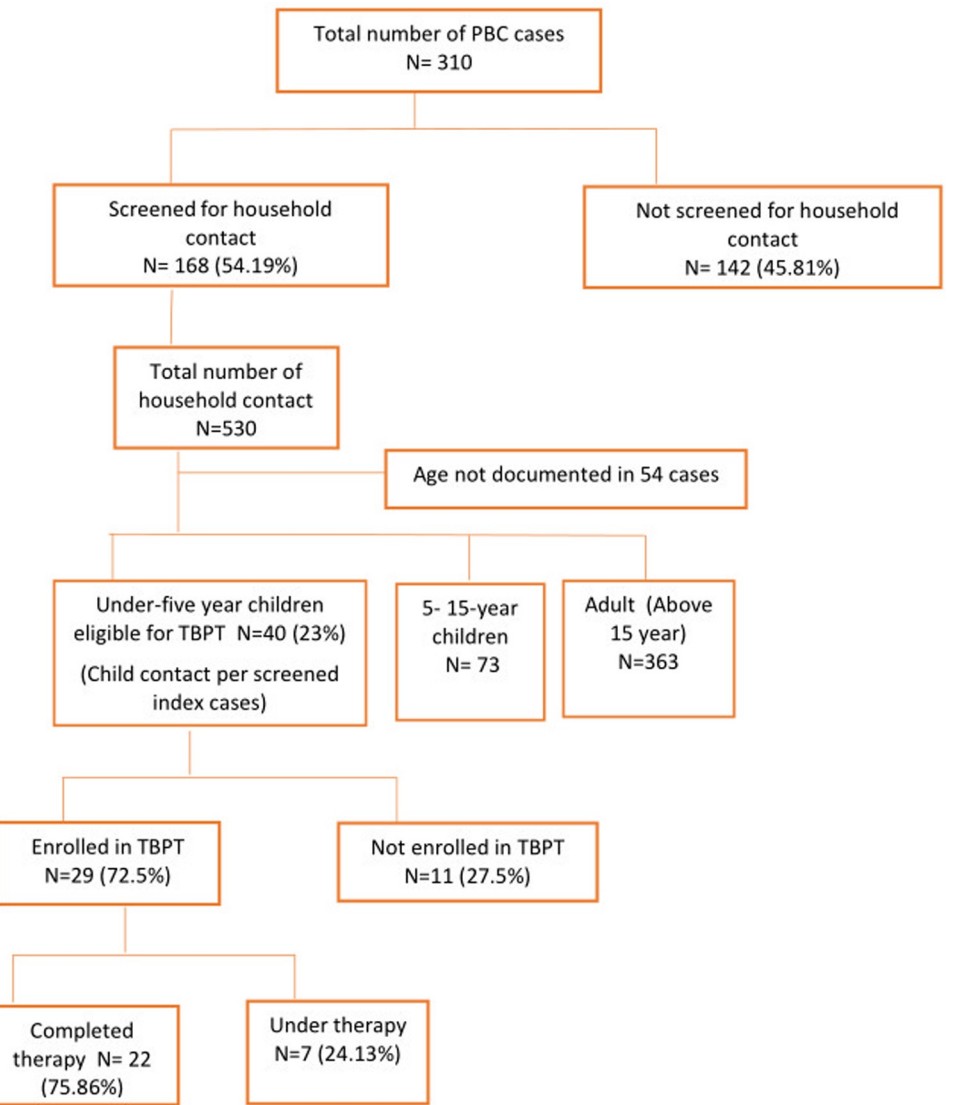

**Fig 1. Contact tracing of PBC cases and TBPT enrolment of under five children.**

*"We identified more than 250 pulmonary positive cases in a year. Moreover, if we screen them well, we can find 110–150 patients who need to be treated with TBPT, but we have given it to only 30–40 patients so far. The follow-up in those patients is poor." (District TB focal person, 56 years, Male)*

**Table 2. Implementation fidelity of quantitative components.**

| Content | Proportion (%) | Implementation Fidelity |
|---|---|---|
| Eligible children initiated TBPT | 72.50 | Moderate |
| Completion of TBPT | 75.86 | Moderate |
| Initiation of TBPT within two months | 86.20 | High |
| Contact tracing for household contact | 54.19 | Moderate |
| Contact screening within two months | 82.73 | High |

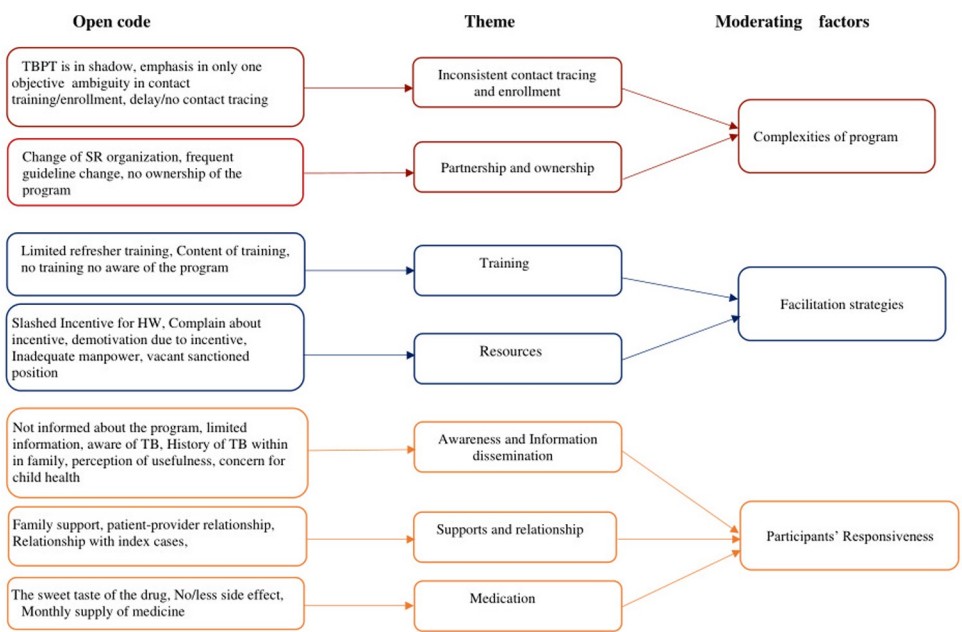

**Fig 2. Moderating factors for the implementation of TBPT program.**

"*These days NTC has been mostly focused on active case finding which might have put the preventive chemotherapy program in the shade." (District TB focal person, 56 years, Male)*

Delay/no contact tracing was a major problem resulting in limited identification of children eligible for therapy, delayed initiation of therapy and no enrollment. Upon asking with contact tracer, frequent relocation/movement of PBC case or household was also reported as a barrier. Observation of the register showed that children from other health facilities were enrolled in TBPT from RTC's initiation.

*"I only look at TB programs and therefore more close to them. At other health centers, they have to look at different programs. The staff for TB is also less there. They don't give so much emphasis to TB which is why finding children is difficult for them.*" (Health care provider, 42 Years, Female)

Change in partner organization was found to have an impact on the thorough implementation of the program. In Kaski district, SR organization Health Research and Social Development Forum (HERD) had started working at first and Japan-Nepal Health and Tuberculosis Research Association (JANTRA) took it over later. Contact tracers in FGD and TB focal person involved in IDI mentioned that frequent change in SR organization and guideline was hindering the smooth running of the program.

"*It was different with HERD. I worked for 5–6 months then. And then it disappeared. And then JANTRA came asking us to do contact tracing. Then, they asked us to stop it in the middle and again restarted it after some time. It has affected timely contact tracing and enrolment in TBPT.*" (Contact tracer, 42 years, Female)

However, staff working in JANTRA has denied this and mentioned that HERD beforehand circulated messages and informed all the responsible health workers to continue the program

but health workers did not take ownership of the program and some health facilities stopped contact tracing which caused an impact on TBPT program.

*"Sometimes they feel like it is not their program and stop working especially when some changes happen in guideline and supporting organization." (District program officer NGO level staff, 30 years, Female)*

**B. Facilitation strategies.**   Health care providers (HCPs) were disappointed by the limited training and time allocated for TBPT during the training. All health care providers interviewed mentioned that only one staff being trained from each health facility. So other HCPs did not want to work in absence of a trained one. District TB focal persons also acknowledge that the district was not been able to conduct refresher training for many years and some health care provider in rural areas were unaware of the program.

*"We have around 150 of them who have not received any training. We could not conduct refresher training for many years. The system of health staff is messed up" (District TB focal person, 58 years, Male).*

Although some HCPs and contact tracer received training, argued that all contents were not discussed in detail and more emphasized on TB case identification than TBPT.

*"Not much time is given for TBPT discussion. We just touched the topics. It would be better if there was a training regarding contact tracing IPT with full process and norms." (Health care provider, 25 years, female)*

Limited availability of the staff due to the vacant sanctioned positions is the major impediment for the enrollment of eligible children in the therapy. One of the health care providers from Regional Tuberculosis Center (RTC) shared that despite the sanctioned position of a doctor, there was no regular availability of doctors and the position is usually vacant.

*"We have doctors only for short term(on contract basis), who stay for 4- months and then leave this place. It's been nearly six months that not a single doctor is available, a paramedic has been looking after the place." (Health care provider, 42 Years, Female)*

The cost of purchasing medicine was not a problem as it was available free of cost. The provision of travel cost was identified as a motivating factor for parents to enroll their children in therapy.

*"It encourages. We should at least thank the government. I think it's better to have something than nothing." (Aunt of a child, 30 years, Female)*

While recent revision in the guideline has deducted incentives of health workers for contact tracing which demotivated them to work.

*"NGO used to give money for contract tracing but they slashed down it now. I feel really bad about it. I have decided not to do contact tracing anymore." (Health care provider, 42 years, Female)*

**C. Participants responsiveness.**   The parents' understanding on TB and with a history of TB among family members were identified as enabling factors in enrollment of therapy. The

majority of the parents, whose children were enrolled in therapy, were found to have a good understanding of the mode of transmission and prevention strategies of TB.

*"When I was a child, my daddy had TB, it got transmitted to my mother. . . There was not such a program at that time like today. Now, even I got the disease. . . but I think that this TBPT will prevent direct transmission of TB to my nephew." (Aunt of a child, 18 years Female).*

Likewise, parents' and HCPs' perceptions of the usefulness of the therapy had resulted in the enrolment of children. In contrast, the majority of parents, whose children were not enrolled in therapy, reported that they were neither informed nor counseled about TBPT. Information given was more limited to TB disease than TBPT, as their own knowledge regarding the TBPT program was limited.

*"Contact tracer and Health worker counseled on the mode of transmission and inquired about other problems. They did not mention TBPT and I have no idea about giving medicines to the children." Mother of a child, 24 years, Female)*

*"If the doctor had told me to provide medicines to the child, there would have not any problem in taking it." (Father of a child, 36 years, Male)*

Family support and patient-provider relationship play an important role in the decision to enroll their children in therapy. The majority of the parents got support from their families.

*"Everyone was supportive because I, the mother had TB and it was necessary to give medicines to the child for prevention"(Mother of a child enrolled in therapy, 25 years, Female)*

Most of the respondents with their children in therapy were satisfied by the health care provider's behavior. However, parents have not enrolled their children in the therapy were not happy that they were not informed about this program. In addition, one of the SR staff acknowledged the possibility of children enrolling in therapy was high if index cases were from the same family.

*"Sometimes, the index case is not from the same family, the case stays on rent where they come into close contact with the children, in such cases most of the children are left behind unenrolled." (SR staff, 30 years, Female)*

Parents mentioned that the taste of the medicine and the absence of its side effects were crucial factors for the completion of the course. There was not any major adverse reaction reported. However, one child got a minor allergy to the drug. Besides, respondents mentioned that the monthly availability of the drug for children is making it easier in the continuation of therapy.

*"The medicines used to dissolve once put into the mouth. My child liked its taste." (Mother of a child who completed therapy, 23 years, Female).*

## Discussion

We examined the implementation fidelity of Tuberculosis preventive therapy (TBPT) for children under five years of age exposed to bacteriologically confirmed pulmonary tuberculosis in Nepal using the CFIF framework. We did a review of secondary data of the TBPT program which

revealed that the majority of the TBPT components were implemented with a moderate level of fidelity. The moderating factors regarding the implementation of TBPT were the complexities of the program, facilitation strategies and participants' responsiveness. Our study showed that these factors affect the implementation fidelity of the program in interconnected ways.

## Adherence of the TBPT

Our study showed that 72.5% of eligible children were enrolled in therapy. In contrast, a study by Singh et al. in India showed only 22% of eligible children were enrolled in therapy [9]. The completion of therapy was 75.86% in our study which is higher than 19.7% in Pune, India and 65% in Guinea-Bissau [7]. The moderate level of implementation fidelity in completion of the recommended dose in our study may be justified by the provision of medicine free of cost and travel cost.

## Moderating factors in the implementation of TBPT

We found that participants experienced the program as a complex intervention due to difficulties faced in contact tracing and enrollment. The study suggests that the more simple and clear the program is more likely to implement authentically [20]. Our study showed that delay or no contact screening and delay in initiation of therapy were identified as the barriers by parents or caretakers whose children were not enrolled in therapy. A study from India by Belgaumkar et al. too has underlined similar findings where paramedical health workers were not identifying and referring eligible children for TB screening and IPT as per the national TB guidelines [14]. The WHO recommends clinical evaluation of household contacts of SS+ cases for active TB. The two major dimensions of contact screening: first- to identify all contact cases of different age group undiagnosed TB among contacts of an index case and, second to enroll all cases into preventive therapy for contacts without TB disease who are susceptible to having disease following recent infection. The WHO recommends clinical evaluation of household contacts of SS+ cases for active TB [24]. However, in our study, most health care providers and contact tracers were more concerned with active case finding rather than both issues. A systematic review by Fox et al. in 2013 has highlighted socio-structural factors such as the movement of TB patients as a hindrance to parents or caretakers, who want to complete IPT to their children [25]. This was experienced in our study by the health care providers and contact tracers during contact tracing of the PBC cases. The Regional Tuberculosis Center (RTC), located in the study area is the medical hub for seeking TB care, in western Nepal. People from different areas come here and temporarily stay in rent for seeking health services. The fear of being stigmatized by house owners and the community has been a primary factor why TB patients don't want to cooperate in contact tracing resulting in no or delay in contact tracing. This is consistent with a study conducted in Rwanda [16]. Our finding highlighted the challenges in terms of stakeholders' ownership of the program, which has affected the implementation of the TBPT program. We noted that there is some role for the implementer SR organization, however, the main ownership of the TBPT program has to be borne by the government. Every stakeholder needs to know the program objectives, their roles and the challenges. The prime stakeholder (i.e. the government) should take ownership of the highest possible level of implementation of the TBPT program.

The facilitation strategies mentioned in the study were found to have both positive and negative influences in the implementation of the program. In our study, limited training opportunity was identified as a challenge, which resulted in inadequate information of health worker. However, an implementation fidelity study by Compaoré et al. showed that better training has increased in the performance of the service provider [26]. In our implementation setting, the government was providing travel costs to the children enrolled in therapy as a motivation.

Such practice is rarely seen in other countries. This could be the reason why the travel cost was not reported as a problem in Nepal. In contrast, one study has reported the travel cost is expensive than the cost of medicine [16]. On the other hand, our study showed that health workers were demotivated by the slash down in the incentive. So, the sustainability of the incentive is a challenging issue, which requires strong commitment and execution by the government.

Participants' responsiveness was observed in terms of awareness, dissemination of information, supports and relationship and medication. Parents or caregiver's understanding of TB diseases, belief on TBPT, their experiences and history of TB among immediate family were identified as enabling factors in the initiation and completion of TBPT to their children. The insecurity of parents or caretakers that their children might get TB was encouraging them to enroll their children in TBPT. This is consistent with studies in Indonesia and Rwanda where parents or caretakers with good understanding and experiences were identified as having good adherence [11, 15]. Inadequate knowledge among all health care providers regarding TBT was responsible for no or delay in the dissemination of information. This is consistent with a study in Indonesia where health workers were having difficulties to educate caregivers about the rationale and need for IPT when the child is asymptomatic [11]. Another key enabler in the initiation and completion of the full course of therapy is the sweet taste of the drug. However, in the study by Triasih et al. in Indonesia, parents were having problems in drug administration due to the bitter taste of the drug [11]. Our study found that side effect has a great impact in parents or caretakers psychology. All of the parents were continuing therapy for their children, as no severe signs and symptoms were observed in children which is similar to a study by Rutherford et al. in Indonesia where the absence of signs and symptoms was identified as a facilitator [15]. In contrast, in Guinea Bissau, despite the low adverse effect of drugs there was a problem in the completion of IPT due to migration and traveling [7].

The strength of out is the mixed-method (Sequential Explanatory) design. It provides quantitative evidence supported by qualitative results with several programmatic implications. This study has some limitations as well. The interview was carried out among parents or caretakers whose children already completed medication which might cause recall bias and to manage this issue, children in therapy were also taken. It is based on secondary data from the TB register, contact tracing and IPT register. So, the study could not account number of contact children among PBC cases not screened for household contact and their perception, so we recommend more research to be carried among them.

## Conclusions

The majority of components of the TBPT program were found to be implemented with a moderate level of implementation fidelity. In assessing moderating factors, delay in contact tracing, lack of ownership towards the program, limited training, and deduction in incentive have made the implementation of the program challenging. Such findings emphasized the need for the health care provider and implementer initiated a sensitization program (i.e. training, orientation, and onsite coaching) and incentive sustainability. However, parents or caretakers knowledge, family supports, availability of travel cost and drug characteristics were seen as beneficial for the initiation and completion of the therapy. These achievements need to be sustained for further improvement of the program.

## Supporting information

**S1 Table. Socio-demographic characteristics of the participants involved in qualitative study.**
(DOCX)

**S1 File. Secondary data extraction tool.**
(DOCX)

**S2 File. In-depth interview and focus group discussion interview guide.**
(DOCX)

**S3 File. Dataset.**
(XLSX)

## Acknowledgments

We are grateful to the National Tuberculosis Center, Bhaktapur, Nepal for permitting us to conduct a study in this area. We would like to extend our gratitude to Dr. Suvesh Kumar Shrestha from Save the Children International and JANTRA team for supporting us during the research. We are thankful to all the participants who participated in the research.

## Author Contributions

**Conceptualization:** Ashmita Ghimire.

**Data curation:** Ashmita Ghimire, Adi Utarini.

**Formal analysis:** Ashmita Ghimire.

**Funding acquisition:** Ashmita Ghimire.

**Investigation:** Ashmita Ghimire, Sagun Paudel, Chhoden Lama Yonzon, Sushmita Sharma.

**Methodology:** Ashmita Ghimire, Adi Utarini.

**Project administration:** Ashmita Ghimire.

**Resources:** Ashmita Ghimire.

**Software:** Ashmita Ghimire.

**Supervision:** Yodi Mahendradhata, Bhuvan K. C.

**Validation:** Ashmita Ghimire, Sagun Paudel, Chhoden Lama Yonzon.

**Visualization:** Ashmita Ghimire.

**Writing – original draft:** Ashmita Ghimire, Yodi Mahendradhata, Sagun Paudel, Chhoden Lama Yonzon, Bhuvan K. C., Sushmita Sharma, Adi Utarini.

**Writing – review & editing:** Ashmita Ghimire, Yodi Mahendradhata, Sagun Paudel, Chhoden Lama Yonzon, Bhuvan K. C., Sushmita Sharma, Adi Utarini.

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
