## [Decision Letter · Decision Letter 0]

1 Dec 2021

PONE-D-21-10469

Implementation fidelity of Tuberculosis Preventive Therapy for under five children exposed to Sputum Smear Positive Pulmonary Tuberculosis in Kaski district, Nepal: an implementation research

PLOS ONE

Dear Dr. Ghimire,

Thank you for submitting your manuscript to PLOS ONE. After careful consideration, we feel that it has merit but does not fully meet PLOS ONE’s publication criteria as it currently stands. Therefore, we invite you to submit a revised version of the manuscript that addresses the points raised during the review process.

We look forward to receiving your revised manuscript.

Kind regards,

Troy D. Moon, M.D., M.P.H

Academic Editor

PLOS ONE

Additional Editor Comments (if provided):

Overall I feel this is an important piece of work and based on my review of the manuscript feel the implementation of the study was done well.

Major issues: This manuscript requires significant editing for english language, flow, and formatting consistency. In its current state the reader must read a paragraph multiple times to determine what the authors were trying to convey.

Minor revisions needed:

Introduction

Lines 57-62. Multiple sentences basically saying the same thing. Please rewrite and consolidate.

Line 96. It is not clear what you mean by a sub-recipient approach

The introduction could potentially be reduced by half its word count without losing its meaning with good editing.

Methods

The framework you use has five domains for measurement: 1) adherence; 2) comprehensiveness of policy; 3) facilitation strategies; 4) quality of delivery; and 5) patient responsiveness. However, you then describe them in terms of 1) adherence and 2) moderating factors without really explaining why you are aggregating domains 2-5 under one term. Consider explaining better or describing your assessment in terms of each domain listed

Results

• Consider combining Tables 1 and 2

• You have two Table #1’s

• For Comprehensiveness of policy be consistent. You also refer to it as Complexity of policy

• Lines 229 – 235. These quotes reinforce that the program was not implemented well but they do not the prior paragraph in which you state the objectives of contact tracing were not understood

Journal Requirements:

2. Please provide the names of the six different health facilities in this study.

3. In your ethics statement in the Methods section and in the online submission form, please provide additional information about the data used in the retrospective part of your study. Specifically, please ensure that you have discussed whether all data were fully anonymized before you accessed them and/or whether the IRB or ethics committee waived the requirement for informed consent. If patients provided informed written consent to have data from their medical records used in research, please include this information.

4. Please provide additional details regarding participant consent for the qualitative research part of your study. In the ethics statement in the Methods and online submission information, please ensure that you have specified whether consent was informed.

7. Thank you for stating the following financial disclosure: 

"This research was a part of postgraduate thesis. The scholarship for this degree was funded by the Special Programme for Research and Training in Tropical Diseases at the World Health Organization (WHO/TDR) and Universitas Gadjah Mada, Indonesia.The funders had no role in study design, data collection and analysis, decision to publish, or preparation of the manuscript."

We note that one or more of the authors is affiliated with the funding organization, indicating the funder may have had some role in the design, data collection, analysis or preparation of your manuscript for publication; in other words, the funder played an indirect role through the participation of the co-authors. If the funding organization did not play a role in the study design, data collection and analysis, decision to publish, or preparation of the manuscript and only provided financial support in the form of authors' salaries and/or research materials, please do the following:

a. Review your statements relating to the author contributions, and ensure you have specifically and accurately indicated the role(s) that these authors had in your study. These amendments should be made in the online form.

b. Confirm in your cover letter that you agree with the following statement, and we will change the online submission form on your behalf: 

“The funder provided support in the form of salaries for authors [insert relevant initials], but did not have any additional role in the study design, data collection and analysis, decision to publish, or preparation of the manuscript. The specific roles of these authors are articulated in the ‘author contributions’ section.

8. Please upload a new copy of Figure 2 as the detail is not clear. Please follow the link for more information: " ext-link-type="uri" xlink:type="simple">https://blogs.plos.org/plos/2019/06/looking-good-tips-for-creating-your-plos-figures-graphics/"
" ext-link-type="uri" xlink:type="simple">https://blogs.plos.org/plos/2019/06/looking-good-tips-for-creating-your-plos-figures-graphics/"

Reviewers' comments:

Reviewer's Responses to Questions

**Comments to the Author**

1. Is the manuscript technically sound, and do the data support the conclusions?

Reviewer #1: Yes

2. Has the statistical analysis been performed appropriately and rigorously? 

Reviewer #1: N/A

3. Have the authors made all data underlying the findings in their manuscript fully available?

Reviewer #1: No

4. Is the manuscript presented in an intelligible fashion and written in standard English?

Reviewer #1: No

5. Review Comments to the Author

Reviewer #1: This manuscript addresses a very interesting issue, which has not been the object of many studies up to date. In 2021, Nepal has joined the list of high MDR/RR-TB burden countries, and the information reported in this manuscript may be very useful for implementers working on the prevention of pediatric tuberculosis. In general, paper’ structure is adequate, and data has been collected and handled properly. Results look reasonably drawn from the data available, and the authors correctly point out limitations, which are common in retrospective studies. However, language used is not very clear, making a good part of the document difficult to follow. I strongly advise the authors to work with a text editor or a writing coach in order to improve text readability.

Nevertheless, because of the importance of this topic, I believe the authors should be given an opportunity to fully review their manuscript and resubmit a second version for evaluation.

The “Introduction” section is well structured and refers to relevant literature. However, language is poor, with many iterations and information repeated in different places. I believe that addressing language issues may result in a quite informative introduction.

In the “Material and methods” section, the sub-section “A conceptual framework for the study” contains some inconsistencies. For example, authors talk about “…comprehensiveness of policy…”, but later this is transformed to “…complexities of policy…” without giving any explanation. The section is well structured, although it would greatly benefit of a review by a text editor.

The “Results” section looks well organized. However, the sub-section “Data sources” has a paragraph repeated. Also, table numbering is incorrect, there are two different “Table 1”. Statistics consist of very simple descriptive parameters and they look alright. Sub-section on “Moderating factors” is really interesting, but again an English language review would greatly improve it.

The “Discussion” and “Conclusion” sections have both a good structure, but again, the language used makes it difficult to read. Main points drawn out seem to be well supported by the data presented in the manuscript.

6. PLOS authors have the option to publish the peer review history of their article (what does this mean?). If published, this will include your full peer review and any attached files.

Reviewer #1: **Yes: **Emilio Jose Valverde

---

## [Author Response · Author response to Decision Letter 0]

15 Jan 2022

Author’s response to reviews

Title: Implementation fidelity of Tuberculosis Preventive Therapy for under five children exposed to Sputum Smear Positive Pulmonary Tuberculosis in Kaski district, Nepal: an implementation research

Authors:

Ashmita Ghimire (ashmi.gr8@gmail.com) 

Yodi Mahendradhata (yodi_mahendradhata@ugm.ac.id) 

 Sagun Paudel (mail4sagun@gmail.com) 

 Chhoden Lama Yonzan (yonzon.chhoden.cy@gmail.com)

 Bhuvan KC (kcbhuvan@gmail.com)

Sushmita Sharma (sushmitaghr@gmail.com)

Adi Utarini (adiutarini@ugm.ac.id)

To,

Troy D. Moon, M.D., M.P.H

Academic Editor

PLOS ONE

Subject: Re-submission of the Revision required revision of manuscript PONE-D-21-10469

Respected Sir/Madam

Thank you for the opportunity to revise our manuscript Implementation fidelity of Tuberculosis Preventive Therapy for under five children exposed to Sputum Smear Positive Pulmonary Tuberculosis in Kaski district, Nepal: an implementation research [PONE-D-21-10469] We are thankful to, editor, and reviewer for the constructive comments and feedback on our manuscript. 

We feel that the inputs received from reviewers are insightful and helped us in improving the manuscript. We have made attempt to fully address these comments and incorporate the feedback in the revised manuscript and believe our revised manuscript represents a significant improvement over our initial submission. 

Regarding the query role of co author and with funding organization I agree with the statement “The funder provided support in the form of salaries for authors [insert relevant initials], but did not have any additional role in the study design, data collection and analysis, decision to publish, or preparation of the manuscript. The specific roles of these authors are articulated in the ‘author contributions’ section.”

Please find attached a response to reviewers for your kind consideration. We hope that you find our responses satisfactory. In addition, I would like to kindly inform you that the minimal data sets for this manuscript are also attached in supporting files.

Sincerely,

Authors

Ashmita Ghimire, MPH

 

Introduction

Lines 57-62. Multiple sentences basically saying the same thing. Please rewrite and consolidate

Response: This has been revised. 

Line 96. It is not clear what you mean by a sub-recipient approach

The introduction could potentially be reduced by half its word count without losing its meaning with good editing.

Response: Briefly define recipient organization

Methods

The framework you use has five domains for measurement: 1) adherence; 2) comprehensiveness of policy; 3) facilitation strategies; 4) quality of delivery; and 5) patient responsiveness. However, you then describe them in terms of 1) adherence and 2) moderating factors without really explaining why you are aggregating domains 2-5 under one term. Consider explaining better or describing your assessment in terms of each domain listed

Response: Implementation fidelity framework has five major domain where is Adherence is itself separate component section, whereas as remaining four comprehensiveness of policy, facilitation strategies, quality of delivery and patient responsiveness comes under moderating factor. Furthermore, quantitative part of our study is addressed by adherence and qualitative section by moderating factor. So we decided to describe in only in two part. 

Results

• Consider combining Tables 1 and 2

• You have two Table #1’s

• For Comprehensiveness of policy be consistent. You also refer to it as Complexity of policy

• Lines 229 – 235. These quotes reinforce that the program was not implemented well but they do not the prior paragraph in which you state the objectives of contact tracing were not understood

Response: Complexity of policy has been changed to comprehensiveness of policy along with some edition in paragraph. 

Response: Edited

Journal Requirements:

Response : Done 

2. Please provide the names of the six different health facilities in this study.

Response : Added in the manuscript. 

3. In your ethics statement in the Methods section and in the online submission form, please provide additional information about the data used in the retrospective part of your study. Specifically, please ensure that you have discussed whether all data were fully anonymized before you accessed them and/or whether the IRB or ethics committee waived the requirement for informed consent. If patients provided informed written consent to have data from their medical records used in research, please include this information.

Response: 

4. Please provide additional details regarding participant consent for the qualitative research part of your study. In the ethics statement in the Methods and online submission information, please ensure that you have specified whether consent was informed.

Response: Make some edition in ethic section

Response: Done

Response: Done 

7. Thank you for stating the following financial disclosure: 

"This research was a part of postgraduate thesis. The scholarship for this degree was funded by the Special Programme for Research and Training in Tropical Diseases at the World Health Organization (WHO/TDR) and Universitas Gadjah Mada, Indonesia.The funders had no role in study design, data collection and analysis, decision to publish, or preparation of the manuscript."

We note that one or more of the authors is affiliated with the funding organization, indicating the funder may have had some role in the design, data collection, analysis or preparation of your manuscript for publication; in other words, the funder played an indirect role through the participation of the co-authors. If the funding organization did not play a role in the study design, data collection and analysis, decision to publish, or preparation of the manuscript and only provided financial support in the form of authors' salaries and/or research materials, please do the following:

a. Review your statements relating to the author contributions, and ensure you have specifically and accurately indicated the role(s) that these authors had in your study. These amendments should be made in the online form.

b. Confirm in your cover letter that you agree with the following statement, and we will change the online submission form on your behalf: 

“The funder provided support in the form of salaries for authors [insert relevant initials], but did not have any additional role in the study design, data collection and analysis, decision to publish, or preparation of the manuscript. The specific roles of these authors are articulated in the ‘author contributions’ section.

Response : The funder provided support in the form of salaries for authors [insert relevant initials], but did not have any additional role in the study design, data collection and analysis, decision to publish, or preparation of the manuscript. The specific roles of these authors are articulated in the ‘author contributions’ section.

8. Please upload a new copy of Figure 2 as the detail is not clear. Please follow the link for more information: https://blogs.plos.org/plos/2019/06/looking-good-tips-for-creating-your-plos-figures-graphics/" https://blogs.plos.org/plos/2019/06/looking-good-tips-for-creating-your-plos-figures-graphics/"

Response : Uploaded

---

## [Editor Report · Decision Letter 1]

2 Feb 2022

Implementation fidelity of Tuberculosis Preventive Therapy for under five children exposed to Sputum Smear Positive Pulmonary Tuberculosis in Kaski district, Nepal: an implementation research

PONE-D-21-10469R1

Dear Dr. Ghimire,

We’re pleased to inform you that your manuscript has been judged scientifically suitable for publication and will be formally accepted for publication once it meets all outstanding technical requirements.

Kind regards,

Troy D. Moon, M.D., M.P.H

Academic Editor

PLOS ONE

---

## [Editor Report · Acceptance letter]

7 Feb 2022

PONE-D-21-10469R1 

Implementation fidelity of Tuberculosis Preventive Therapy for under five children exposed to Sputum Smear Positive Pulmonary Tuberculosis in Kaski district, Nepal: an implementation research 

Dear Dr. Ghimire:

I'm pleased to inform you that your manuscript has been deemed suitable for publication in PLOS ONE. Congratulations! Your manuscript is now with our production department. 

Kind regards, 

on behalf of

Dr. Troy D. Moon 

Academic Editor

PLOS ONE